# Kinetic resolution of substituted amido[2.2] paracyclophanes via asymmetric electrophilic amination

Shaoze Yu[1], Hanyang Bao[1], Dekun Zhang[1] & Xiaoyu Yang [1]✉

Planar chiral [2.2]paracyclophane derivatives are a type of structurally intriguing and practically useful chiral molecules, which have found a range of important applications in the field of asymmetric catalysis and material science. However, access to enantioenriched [2.2]paracyclophanes represents a longstanding challenge in organic synthesis due to their unique structures, which are still highly dependent on the chiral chromatography separation technique and classical chemical resolution strategy to date. In this work, we report on an efficient and versatile kinetic resolution protocol for various substituted amido[2.2]paracyclophanes, including those with *pseudo-geminal*, *pseudo-ortho*, *pseudo-meta* and *pseudo-para* disubstitutions, using chiral phosphoric acid (CPA)-catalyzed asymmetric amination reaction, which was also applicable to the enantioselective desymmetrization of an achiral diamido[2.2]paracyclophane. Detailed experimental studies shed light on a new reaction mechanism for the electrophilic aromatic C-H amination, which proceeded through sequential triazane formation and N[1,5]-rearrangement. The facile large-scale kinetic resolution reaction and diverse derivatizations of both the recovered chiral starting materials and the C-H amination products showcased the potential of this method.

Since the first discovery of [2.2]paracyclophane (PCP) by gas-phase pyrolysis of *para*-xylene[1], as well as the practical synthesis through an intramolecular cyclization route[2] around 1950, these structurally interesting molecules have drawn considerable research interests due to their unique photophysical and optoelectronic properties[3]. Moreover, the substituted [2.2]paracyclophanes displayed intriguing planar chirality, and these chiral skeletons have been widely studied in the development of chiral catalysts and ligands, as well as versatile chiral synthons in material science[4–7] (Fig. 1a). Specifically, the amino[2.2]paracyclophanes represent as an important type of substituted [2.2]paracyclophanes, which have found numerous applications in various fields, such as the development of planar chiral bisthiourea catalyst[8], N-heterocyclic carbene (NHC) ligands[9], thermally activated delayed fluorescence (TADF) circularly polarized luminescence (CPL) emitters[10] and others[11–20].

However, the access to enantiomerically pure [2.2]paracyclophanes represents a formidable challenge in organic synthesis[7,21], which still largely relied on the chiral chromatography separation or classical chemical resolution by forming diastereomeric salts or covalent adducts with chiral reagents[22–25]. On the other hand, catalytic kinetic resolution (KR) has emerged as an attractive and practical approach to afford enantioenriched molecules, which have found widespread applications in both academia and industry[26–29]. In the past two decades, a number of elegant catalytic kinetic resolution methods have been developed to access chiral PCP derivatives, including the asymmetric C-N coupling of PCP-based dihalides[30], hydrolysis/esterification of PCP-based diphenol derivatives[31,32], reduction of the PCP-based aldehydes and ketones[33–37], and reduction/addition of PCP-based imines[38–40] (Fig. 1b). Despite these advances, most of these methods still suffer certain limitations, including unsatisfactory KR performances, limited substrate scope, and the inconvenience of accessing both enantiomers of the planar chiral PCP derivatives. Moreover, to the best of our knowledge, the catalytic kinetic

[1]School of Physical Science and Technology, ShanghaiTech University, Shanghai 201210, China. ✉e-mail: yangxy1@shanghaitech.edu.cn

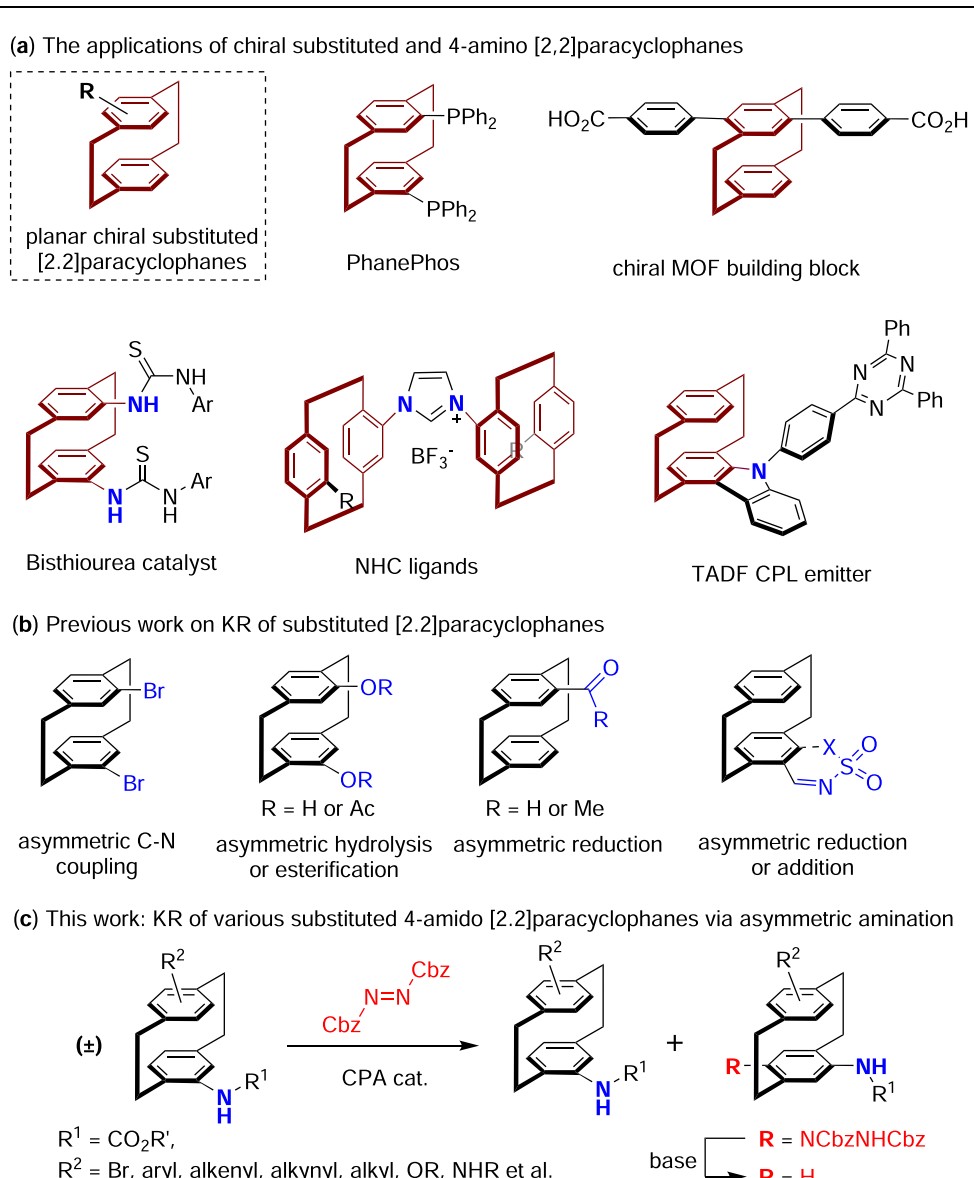

**(a)** The applications of chiral substituted and 4-amino [2,2]paracyclophanes

planar chiral substituted [2.2]paracyclophanes

PhanePhos

chiral MOF building block

Bisthiourea catalyst

NHC ligands

TADF CPL emitter

**(b)** Previous work on KR of substituted [2.2]paracyclophanes

asymmetric C-N coupling

R = H or Ac
asymmetric hydrolysis or esterification

R = H or Me
asymmetric reduction

asymmetric reduction or addition

**(c)** This work: KR of various substituted 4-amido [2.2]paracyclophanes via asymmetric amination

R¹ = CO₂R',
R² = Br, aryl, alkenyl, alkynyl, alkyl, OR, NHR et al.

base
R = NCbzNHCbz
R = H

**Fig. 1 | Kinetic resolution of substituted [2.2]paracyclophanes. a** The applications of chiral substituted and 4-amino [2,2]paracyclophanes. **b** Previous work on KR of substituted [2.2]paracyclophanes. **c** This work: KR of various substituted 4-amido [2.2]paracyclophanes via asymmetric amination.

resolution of amino[2.2]paracyclophanes is currently unprecedented, and these valuable chiral skeletons could only be afforded by chemical resolution[22] or the derivatizations from other known chiral PCP derivatives[11,20].

Recently, our group has developed a highly efficient kinetic resolution method for arylamines bearing central[41–43] or axial chirality[44,45], by the utilization of chiral phosphoric acid[46–49] (CPA) catalyzed asymmetric amination reaction with commercially available azodicarboxylates[50,51]. Herein, we report on the efficient and versatile KR of a variety of substituted amido[2.2]paracyclophanes, including those with *pseudo-geminal, pseudo-ortho, pseudo-meta* and *pseudo-para* disubstitutions, through the CPA-catalyzed asymmetric amination protocol, in which both enantiomers of the substituted amido[2.2]paracyclophanes could be readily afforded by the facile derivatization of the C-H amination products (Fig. 1c).

## Results and discussion
### Reaction condition optimizations
We began our study by selecting the racemic 4-NHBoc substituted [2.2]paracyclophane **1a** as the model substrate, which could be easily

prepared from the Pd-catalyzed C-N coupling of 4-bromo substituted PCP and BocNH₂ (Table 1). Encouragingly, the KR of racemic **1a** with dibenzyl azodicarboxylate **2a** (1.0 equiv.) in the presence of CPA **A1** (10 mol%) in toluene at 20 °C afforded the desired C-H amination product **3a** with 88% ee and the recovered **1a** with 32% ee, corresponding to a conversion of 27% and selectivity factor[52] (s) of 21 (entry 1). Next, a series of CPA catalysts bearing various *ortho*-substitutions and varied chiral scaffolds were screened in this reaction (entries 2–8), and we were pleased to find that the BINOL-derived CPA catalyst **A5** bearing *ortho*-(9-phenanthracenyl)-substituents could generate the C-H amination product **3a** in 94% ee and the recovered **1a** with 90% ee (s = 100, entry 5). It is worth mentioning that the sterically hindered CPA could not provide the desired C-H amination product under the same conditions (entries 6 and 8). A diverse set of solvents suitable for solubilizing substrate **1a** was examined, which suggested that the nonpolar solvents gave more superior KR performances (entries 9–13). Satisfyingly, the KR reaction in CCl₄ provided **3a** with 95% ee and the recovered **1a** with 90% ee, corresponding to a selectivity factor of 120 (entry 13). Moreover, it is noteworthy that the reaction conducted in CCl₄ exhibited the highest reaction rate, which would enable a

**Table 1 | Optimization of reaction conditions for KR of amido[2.2]paracyclophane[a]**

A1, BINOL, Ar = 1-naphthyl
A2, BINOL, Ar = 2-naphthyl
A3, BINOL, Ar = 1-pyrenyl
A4, BINOL, Ar = 9-anthracenyl
A5, BINOL, Ar = 9-phenanthracenyl
A6, BINOL, Ar = 2,4,6-(iPr)$_3$C$_6$H$_2$
A7, H8-BINOL, Ar = 9-anthracenyl
A8, Ar = 9-anthracenyl

| Entry | Cat. | Sol. | ee$_s$ (%)[b] | ee$_p$ (%)[b] | Conv. (%)[c] | s[d] |
|---|---|---|---|---|---|---|
| 1 | A1 | Toluene | 32 | 88 | 27 | 21 |
| 2 | A2 | Toluene | 28 | 56 | 33 | 4.6 |
| 3 | A3 | Toluene | 51 | 83 | 38 | 18 |
| 4 | A4 | Toluene | 84 | 94 | 47 | 86 |
| 5 | A5 | Toluene | 90 | 94 | 49 | 100 |
| 6 | A6 | Toluene | ~0 | ND | Trace | ND |
| 7 | A7 | Toluene | 55 | 96 | 36 | 85 |
| 8 | A8 | Toluene | ~0 | ND | Trace | ND |
| 9 | A5 | DCM | 45 | 87 | 34 | 22 |
| 10 | A5 | CHCl$_3$ | 52 | 77 | 40 | 13 |
| 11 | A5 | Et$_2$O | ~0 | ND | Trace | ND |
| 12 | A5 | Benzene | 83 | 94 | 47 | 84 |
| 13 | A5 | CCl$_4$ | 90 | 95 | 49 | 120 |
| 14[e] | A5 | CCl$_4$ | 99 | 93 | 52 | 145 |
| 15[f] | A5 | CCl$_4$ | 96 | 96 | 50 | 194 |
| 16[g] | A5 | CCl$_4$ | 79 | 97 | 45 | 159 |
| 17[h] | A5 | CCl$_4$ | 98 | 93 | 51 | 127 |
| 18[e] | A5 | Toluene | 88 | 95 | 48 | 114 |
| 19[f] | A5 | Toluene | 90 | 95 | 49 | 121 |

[a]Reactions were run with **1a** (0.05 mmol), **2a** (0.05 mmol) and the CPA catalyst (0.005 mmol, 10 mol%) in solvent (1 ml) with 4 Å molecular sieves (50 mg) at 20 °C.
[b]The ee values were determined by HPLC analysis on a chiral stationary phase.
[c]Conversion (C) = ee$_s$/(ee$_s$ + ee$_p$).
[d]s = ln[(1 − C)(1 − ee$_s$)]/ln[(1 − C)(1 + ee$_s$)].
[e]At 0 °C.
[f]At −10 °C.
[g]At −20 °C.
[h]At −10 °C, **2a** (0.035 mmol) and CCl$_4$ (0.5 ml) was used.

decrease in both the reaction temperature and the necessary amount of azodicarboxylate. Consequently, the reaction temperature was investigated as well (entries 14–16), which suggested that −10 °C was the optimal reaction temperature, and both the recovered **1a** and C-H amination product **3a** could be afforded with 96% ee ($s = 194$, entry 15). Finally, to facilitate the purification process, the amount of azodicarboxylate **2a** was reduced to 0.75 equivalents while increasing the concentration of this reaction, which still provided satisfactory KR performance ($s = 127$, entry 17). Recognizing the potential toxicity associated with CCl$_4$, the reaction was also explored in toluene at lower temperatures (entries 18–19). Although this resulted in slightly diminished KR performance, it nonetheless offered an alternative set of conditions for the KR of amido[2.2]paracyclophanes.

### Substrates scope investigation

With the optimal KR conditions in hand, we set out to investigate the scope of substituted amido[2.2]paracyclophanes (Fig. 2). First, a series of N-substitutions were screened, and we were delighted to find that various N-alkoxycarbonyl groups were well tolerated in this reaction, which could be selectively removed under distinct conditions (**1a**–**1e**). Next, we turned our attention to the disubstituted amido[2.2]paracyclophanes. The *pseudo-geminal*-Br-substituted amido-PCP **1f** was examined under the standard KR conditions, which readily provided the C-H amination product **3f** in 46% yield with 97% ee, as well as the recovered ($S_p$)-**1f** in 51% yield with 89% ee, corresponding to an *s* factor of 198. The absolute configure of recovered **1f** was assigned as $S_p$, which was unambiguously determined by X-ray crystallography analysis, and the configurations of the other amido-PCPs were assigned by analogy to this structure. Additionally, a series of amido-PCPs bearing aryl, alkenyl, alkynyl and alkyl groups at the *pseudo-geminal*-position were all amenable to this method, which provided good to high KR performances (**1g**–**1k**). Notably, the *pseudo-geminal* OAc-substituted amido-PCP **1l** was also a feasible variant under the KR conditions, which was a valuable chiral synthon for the development of chiral catalysts and ligands[11]. A wide range of substitutions at the *pseudo-ortho*-position of the amido-PCPs were also investigated, which were all well compatible with this KR method, giving *s* factors up to 180 (**1m**–**1v**). It is worth mentioning that diverse functional groups were readily feasible at this position, including Br (**1m**), OAc (**1s**), CHO (**1t**), CH$_2$OH (**1u**) and CO$_2$Me (**1v**), which would facilitate the further derivatizations of these chiral PCP building blocks. The *pseudo-meta*- and *pseudo-para*-substituted amido-PCPs bearing bromo, aryl, alkenyl and alkyl groups were studied with the KR method as well, which could give both the recovered SM ($R_p$)-**1** and the C-H amination products **3** up to >95% ee, albeit slighted modified conditions were employed for these substrates (**1w**–**1z** for *pseudo-meta*-substituted amido-PCPs and **1aa**–**1ad** for *pseudo-para*-substituted amido-PCPs). Furthermore, the enantioselective desymmetrization of the achiral *pseudo-para*-substituted diamido-PCP **4a** was studied using this method by modifying the reaction conditions, which gave access to the expected C-H amination product **5a** in 40% yield with 99% ee, with the unreacted **4a** in 23% yield and achiral di-amination product as the major byproducts.

### Mechanistic studies

To shed light on the detailed mechanism for these reactions, a series of control experiments were performed (Fig. 3). The treatment of the N-Boc-N-Me-substituted amido-PCP **6a** with the standard KR conditions failed to provide the desired C-H amination product, which suggested the N-H moiety of the substrate played a crucial role in this reaction (Fig. 3a). Surprisingly, the 2,5-dimethyl-substituted N-Boc aniline **6b**, a simplified noncyclophane-type analogue of **1a**, was also found to be unreactive to the standard KR conditions, suggesting the indispensable role of the [2.2]paracyclophane scaffold in the reactivity of this reaction (Fig. 3b). Additionally, other N-substitution groups apart from the N-alkoxycarbonyl groups in the amido-PCP substrates were also examined in the KR reaction (Fig. 3c). Notably, the amido-PCPs bearing various N-acyl substitutions resulted in dramatically diminished KR performances (**6c**–**6e**, with *s* of 3.8 ~ 9.5), while the N-Ts substituted PCP barely exhibited any KR effect (with both recovered **6f** and the C-H amination product **7f** obtained in racemic form). These results suggested that the N-substitutions of the amido-PCPs play a critical role for the high stereoselectivity of these KR reactions, which may potentially affect the interaction between the CPA catalyst and the amido group. The N-unsubstituted 4-amino[2.2]paracyclophane **6g** was also subjected to examination under the standard conditions, which surprisingly did not yield the expected C-H amination product **7g** after 24 h. Instead, the triazane product **8g** was afforded via the direct addition of the amino group to the azodicarboxylate[44] (Fig. 3d). However, both the triazane product **8g** and the recovered **6g** were obtained with low enantioselectivities. Interestingly, raising the temperature to 20 °C and prolonging the reaction time to 48 h resulted in a mixture of **8g** and the expected C-H amination product **7g**, with poor KR performance. Moreover, during the screening of the substrate scope, a significant amount of triazane product **8** was also generated with high enantioselectivity in the KR of *pseudo-meta*-substituted amido-PCPs when performing the reaction at −10 °C (compared to 0 °C in the substrate scope). This type of intermediate was also observed for the *pseudo-para*-substituted amido-PCP **1ac**, when analyzing the reaction mixture at the early stage of the reaction. For instance, the KR reaction of *pseudo-meta*-Ph-substituted amido-PCP **1x** under the standard conditions (−10 °C, 40 h) provided the recovered ($R_p$)-**1x** in 51% yield with 92% ee and the C-H amination product **3x** in 37% yield with 94%, as well as the triazane **8x** in 8% yield with 95% ee (Fig. 3e). Careful monitoring this reaction by chiral HPLC analysis indicated that the triazane **8x** was actually formed first, which was then slowly converted into the C-H amination product **3x**. To study the transformation between **8x** and **3x**, several control experiments were conducted. In the absence of CPA catalyst, no conversion between **8x** and **3x** was observed, while the enantioenriched **8x** could be readily converted into chiral **3x** with retained enantiopurity in the presence of a racemic phosphoric acid catalyst. Treatment of racemic triazane **8x** with CPA (*R*)-**A5** resulted in the formation of **3x** with 25% ee and the recovered **8x** with 63% ee, corresponding to an *s* factor of 2.9 (Fig. 3f). These results suggested that the CPA-catalyzed rearrangement from **8x** to **3x** was not the primary stereoselective step in the KR reaction. To further elucidate this hydrazine-shift reaction, a cross-over experiment was performed. A 1:1 mixture of racemic triazane **8x** and amido-PCP **1a** was subjected to the standard conditions, which afforded the C-H amination product **3x** in 43% yield with 32% ee and the recovered **8x** in 43% yield with 34% ee ($s = 2.6$, Fig. 3g). Notably, no C-H amination product **3a** derived from **1a** was detected, which implied that the hydrazine-shift reaction proceeded through an intramolecular manner. Based on these results, a plausible reaction mechanism for this KR reaction was proposed (Fig. 3h). The CPA-catalyzed direct addition of the amido group to azodicarboxylate (formation of triazanes **8**) was believed to be the stereodetermining step, in which the ($S_p$)-amido-PCP **1** was the kinetically favored enantiomer due to the dual-hydrogen bonding activation between CPA catalyst with substrates and azodicarboxylate. The acid-promoted hydrazine-shift from the amino group to the *para*-position was proposed to proceed through a *para*-semidine-type rearrangement, involving either a stepwise double N[1,3]-sigmatropic shift[53,54] or a one-step N[1,5]-sigmatropic shift process[55,56]. In most cases, the rearrangement step is faster than the triazane formation step, and the corresponding triazane intermediate cannot be detected. However, due to the steric hindrance around the *para*-position of aniline for *pseudo-meta*- and *pseudo-para*-substituted amido-PCPs,

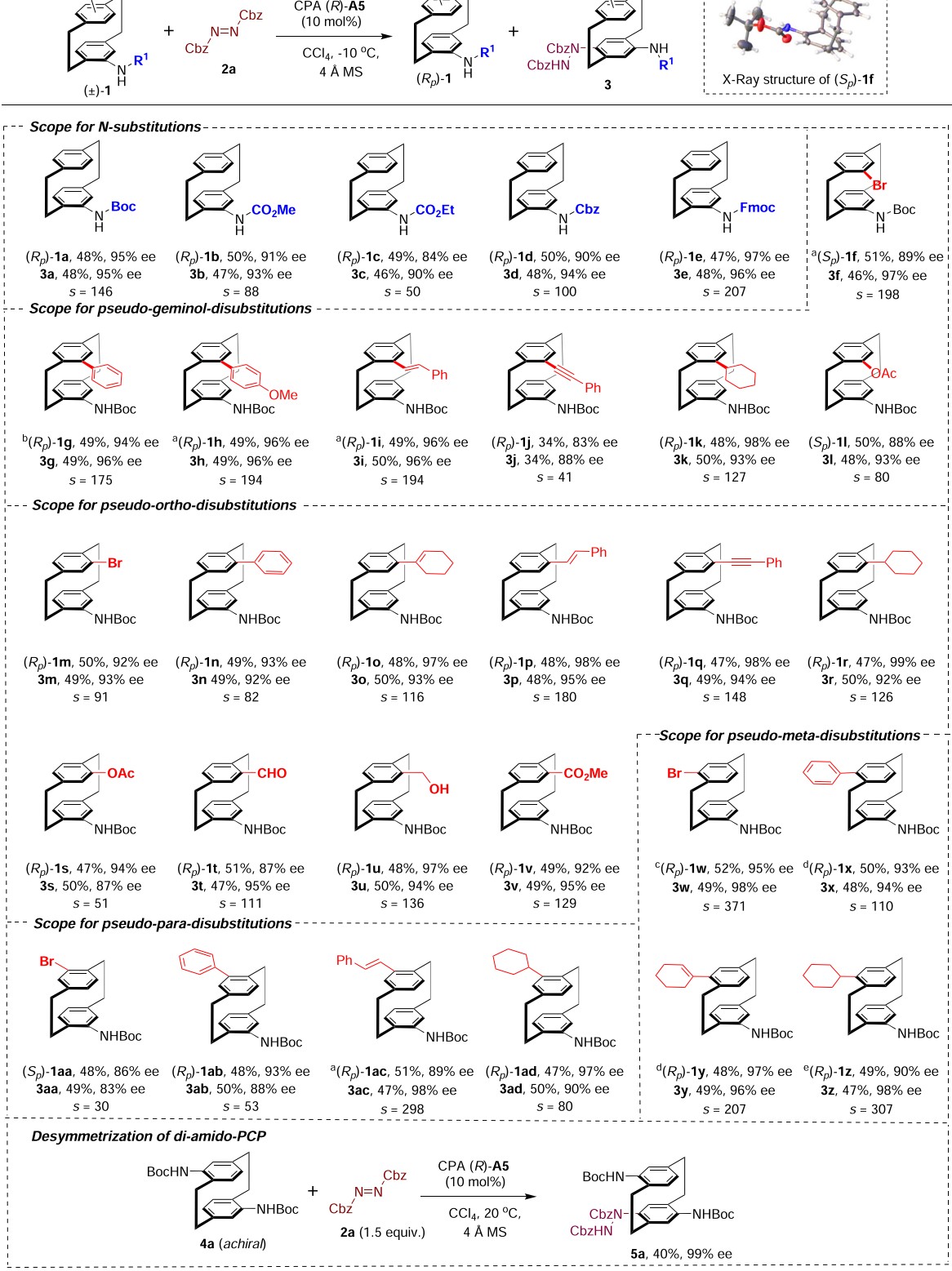

**Fig. 2 | Scope for KR of substituted amido[2.2]paracyclophanes.** Reactions were run with **1** (0.2 mmol), **2a** (0.14 mmol) and the CPA (*R*)-**A5** (0.02 mmol, 10 mol%) in CCl₄ (2 ml) with 4 Å molecular sieves (200 mg) at −10 °C. Yields refer to isolated yields. The ee values were determined by HPLC analysis on a chiral stationary phase. Conversion (C) = ee$_s$/(ee$_s$ + ee$_p$). s = ln[(1 − C)(1 − ee$_s$)]/ln[(1 − C)(1 + ee$_s$)]. Note: (a)

Reactions were performed in toluene at −20 °C. (b) Reactions were performed in toluene at −30 °C. (c) Reactions were performed at 0 °C. (d) Reactions were performed using **2a** (0.65 equiv) at 0 °C. (e) Reactions were performed using **2a** (0.6 equiv) at 0 °C.

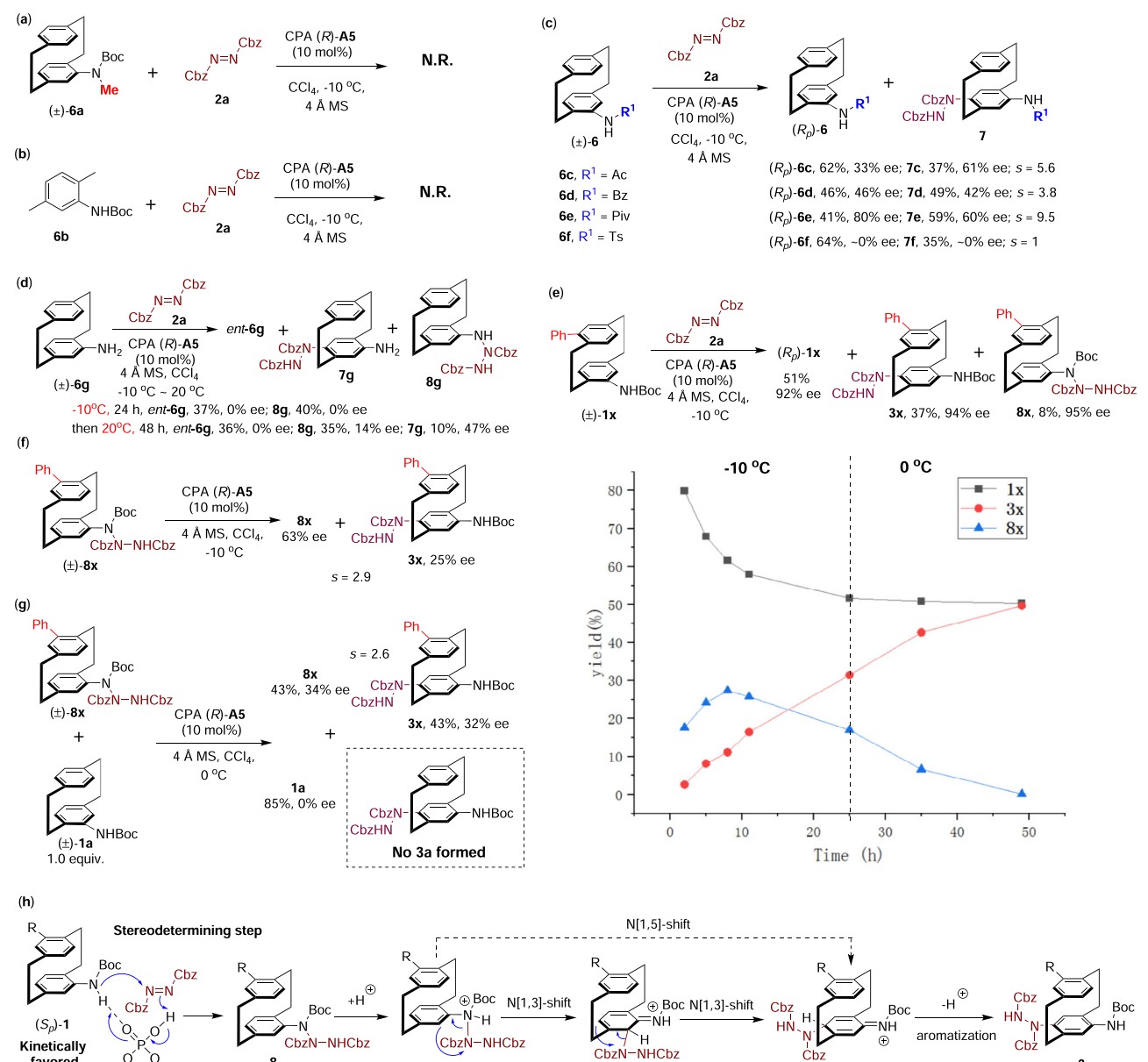

**Fig. 3 | Experimental mechanistic studies and postulated reaction mechanism. a** Control experiment of N-Boc-N-methylated amido-PCP. **b** Control experiment of N-Boc aniline **6b**. **c** KR of amido-PCPs bearing various N-acyl groups. **d** Control experiment of the N-unsubstituted amino-PCP **6g**. **e** Monitoring the reaction of

*pseudo-meta*-substituted amido-PCP **1x** at −10 °C. **f** Reaction of triazane **8x** in the presence of CPA catalysts. **g** Cross-over experiment of triazane **8x** and amido-PCP **1a**. **h** Proposed reaction mechanism.

the rearrangement step is slower than the initial triazane formation step, leading to the detection or isolation of the triazanes **8**.

## Derivatizations of chiral products

To demonstrate the practicability of this method, a large-scale KR of racemic **1a** (2.0 mmol) was performed using the standard conditions, which readily gave access to the C-H amination product **3a** in 51% yield with 94% ee and the recovered ($R_p$)-**1a** in 47% yield with 99% ee, corresponding to an *s* factor of 170 (Fig. 4a). The derivatizations of these planar chiral products were then investigated to showcase the utilities of this method. Firstly, the derivatizations of the C-H amination products were examined. The treatment of **3a** with strong basic conditions at 70 °C led to the facile removal of the hydrazine moiety, and provided the ($S_p$)-**1a** in 94% yield with 94% ee[41] (Fig. 4b). Therefore, both enantiomers of the amido-PCPs could be readily afforded without the need for two KR reactions using opposite enantiomer of the CPA

catalyst. Moreover, the substituted hydrazine group in **3a** was readily converted into a NH₂ group (**9a**) via facile catalytic hydrogenation, and this type of planar chiral diamine scaffold may find diverse applications in organic synthesis and material science. For instance, the coupling of **9a** with isothiocyanate afforded the primary amine-thiourea-type derivative **10a**, which could be further modified into a planar chiral bis-thiourea derivative **11a**. Notably, both of these two products have the potential to serve as chiral organocatalysts in asymmetric catalysis. In addition, treatment of **9a** with NaNO₂ and KI afforded the *para*-iodinated amido-PCPs **12a**, which is a versatile planar chiral building block for further derivatizations as well. Next, the derivatizations of the recovered chiral amido-PCPs were also studied (Fig. 4c). After switching the N-protecting group of the *pseudo-geminal*-Br-substituted amido-PCP ($S_p$)-**1f** from Boc to Piv group, the NHPiv group directed sp²-C−H borylation[57] yielded a functional group-rich PCP derivative **14f** in 80% yield with 91% ee, which bear one amido, one

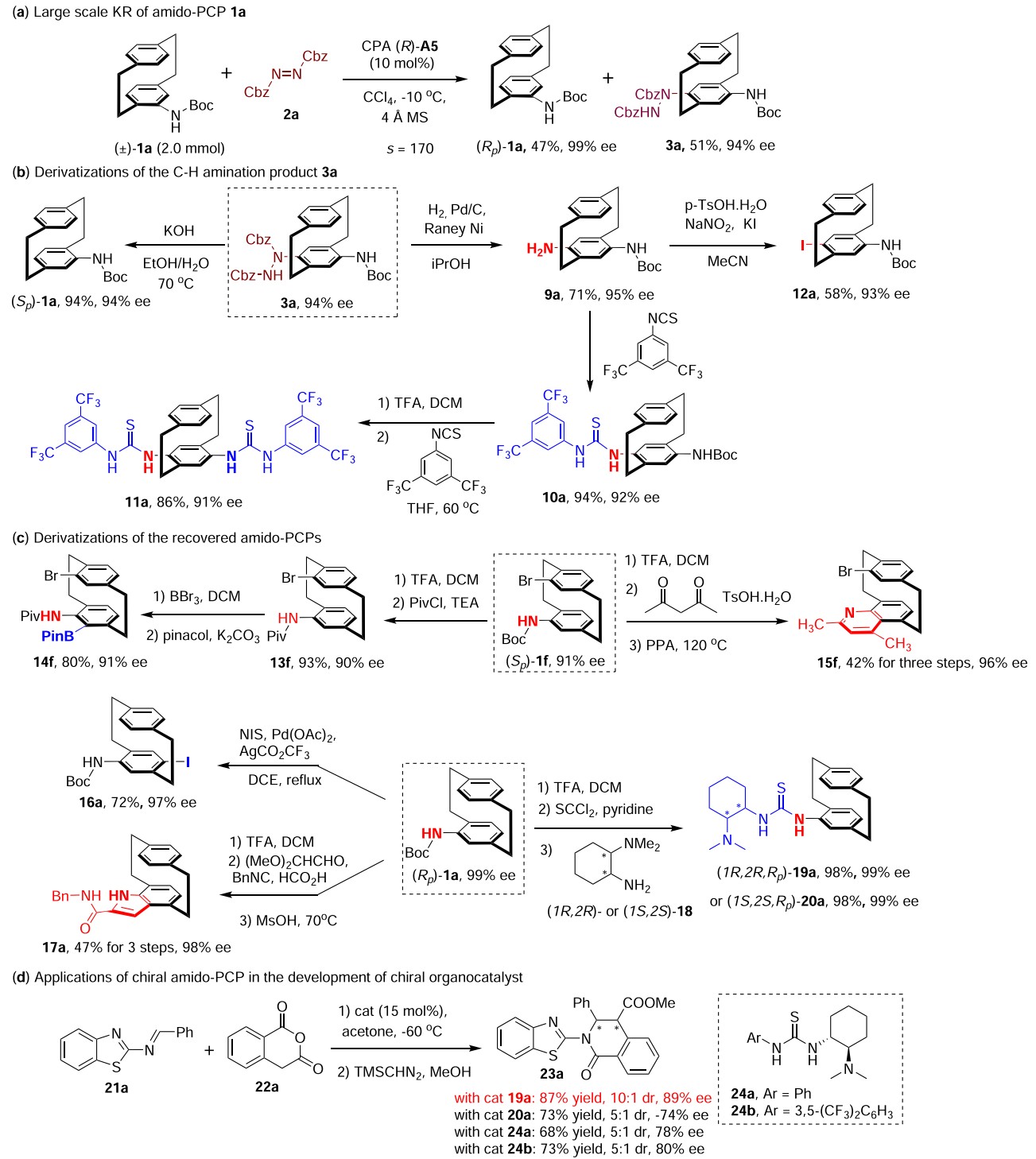

**Fig. 4 | Large scale reaction and derivatizations of the chiral products. a** Large scale KR of amido-PCP **1a**. **b** Derivatizations of the C-H amination product **3a**.
**c** Derivatizations of the recovered amido-PCPs. **d** Applications of chiral amido-PCP in the development of chiral organocatalysts.

bromide and one boronate group. In addition, after removal of the N-Boc group of ($S_p$)-**1f**, the Combes quinoline synthesis was performed with acetyl acetone, yielding the corresponding quinoline-containing PCP derivative **15f**. In addition, the Pd-catalyzed *para*-iodination of ($R_p$)-**1a** with NIS afforded **16a** in 72% yield with 97% ee, which was also the enantiomer of **12a**. Furthermore, after release of the NH₂ group, the four-component reaction of the aniline moiety with glyoxal dimethyl acetal, formic acid, and benzyl isocyanide gave the indole-containing PCP derivative **17a**[58]. The utilization of these planar chiral

amido-PCP skeletons in the development of chiral organocatalysts was explored as well. Sequential coupling of the amino-PCP segment and (1R,2R)- or (1S,2S)-*N,N*-dimethylcyclohexane-1,2-diamine **18** with thio-phosgene (SCCl₂) afforded the diastereomeric tertiary amine-thiourea bifunctional catalysts **19a** and **20a**, respectively. These two products, along with the commercially available chiral tertiary amine-thiourea catalysts (**24a** and **24b**), were screened as chiral bifunctional catalysts in the asymmetric [4 + 2] cyclizations of 2-benzothiazolimine **21a** with homophthalic anhydride **22a**[59] (Fig. 4d). Notably, the reaction

catalyzed by (1R,1S,Rp)-**19a** (15 mol%) yielded the cyclization product **23a** in 87% yield with 10:1 dr and 89% ee, whereas the catalysts with the achiral aniline moiety (**24a** and **24b**) produced the product with erosive yield, diastereoselectivity, and enantioselectivity. A significant match-mismatch effect was observed since the reaction enabled by (1S,2S,Rp)-**20a** generated the opposite enantiomer of product with both diminished dr and ee values.

In conclusion, we have developed an efficient kinetic resolution method for amido-PCPs through the CPA-catalyzed asymmetric C-H amination using commercially available azodicarboxylates. This KR method features a remarkably broad substrate scope, enabling the successful resolution of various *pseudo-geminal-*, *pseudo-ortho-*, *pseudo-meta-* and *pseudo-para*-substituted amido-PCPs with excellent KR performances (with *s* factors up to 371). Moreover, this reaction allows for the enantioselective desymmetrization of an achiral diamido-PCP substrate. Detailed experimental studies unraveled a new reaction mechanism for the electrophilic aromatic C-H amination, involving the sequential triazane formation and N[1,5]-rearrangement process, with the CPA-catalyzed direct addition of the amido group to azodicarboxylate as the stereo-determining step. The feasibility of large-scale KR reaction, along with the potential for diverse derivatizations of both the recovered chiral starting materials and the C-H amination products, particularly the utilization of the planar chiral amino-PCP scaffold in chiral organocatalysts development, further underscore the value of this methodology.

## Methods

### Representative procedure
To a solution of racemic **1** (0.2 mmol, 1.0 equiv.), (R)-**A5** (0.02 mmol, 0.1 equiv.) and activated 4 Å MS (200 mg) in $CCl_4$ or toluene (1 ml) was added a solution of **2a** (0.14 mmol, 0.7 equiv.) in dry $CCl_4$ or toluene (1 ml) at designed temperature under $N_2$ atmosphere. After achieving appropriate conversion as indicated by HPLC analysis of the reaction mixture, the reaction mixture was quenched with $Et_3N$ (20 µl) and concentrated under vacuum to give a residue, which was purified by column chromatography to afford the recovered product **1** and product **3**.

## Data availability
The authors declare that the data supporting the findings of this study are available within the article and Supplementary Information file, or from the corresponding author upon request. The X-ray crystallographic coordinates for structures reported in this study have been deposited at the Cambridge Crystallographic Data Centre (CCDC), under deposition numbers CCDC 2248134 (for (Sp)-**1f**). These data can be obtained free of charge from The Cambridge Crystallographic Data Centre via www.ccdc.cam.ac.uk/data_request/cif.

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

## Acknowledgements

The authors gratefully acknowledge NSFC (grant nos. 22171186, 22222107), Double First-Class Initiative Fund of ShanghaiTech University, and ShanghaiTech University start-up funding for financial support. The authors thank the support from Analytical Instrumentation Center (# SPST-AIC10112914), SPST, ShanghaiTech University and Mr. Huanchao Gu for the X-ray crystallographic analysis.

## Author contributions

S.Y., H.B. and D.Z. performed the experiments. X.Y. directed the project and wrote the paper with the feedback from other authors.

## Competing interests

The authors declare no competing interests.
