## [Peer Review File · Nature Communications]

REVIEWER COMMENTS

Reviewer #1 (Remarks to the Author):

The manuscript of Yang and co-workers describes a highly efficient kinetic resolution of differently substituted amido[2.2]paracyclophanes. This versatile transformation proceeds via an asymmetric electrophilic amination promoted by chiral phosphoric acid catalysts. After the optimization of the reaction conditions, the kinetic resolution protocol was readily applied to a wide variety of racemic substrates. A valuable desymmetrization of a centrosymmetric diamido pCp was also achieved following this strategy. Several control experiments have been conducted by the authors to propose a plausible reaction mechanism. Finally, the utility of the enantioenriched products obtained using this method was showcased by performing useful derivatization reactions. High added value compounds were easily obtained, including an example of a promising organocatalysts.

The work is clearly designed, and the conclusions are consistent with the experimental data. The supporting information appears complete. The proposed transformation grants a straightforward access to enantioenriched nitrogen-containing pCps that can serve as valuable synthetic intermediate for a variety of applications ranging from asymmetric catalysis to material sciences. I believe that the results of this study will be relevant to a broad readership of scientists. I therefore think that this manuscript should be accepted for publication in Nature Communications after addressing the following minor remarks.

- 1) The manuscript is well-structured, but some typos can be found in the text and a few sentences are hard to understand. English needs to be polished.
- 2) When describing the optimization of the reaction conditions, authors should comment on the solubility of the starting materials and products in the solvents employed to perform the kinetic resolutions.
- 3) Authors should more clearly describe the advantages and disadvantages of employing CCl₄ as a solvent instead of toluene.
- 4) Screening of different temperatures for the reaction performed in toluene may be interesting and could be added to the optimization table.
- 5) In figure 2, when describing the desymmetrization reaction of the meso compound 4a, authors should report more precisely the yield of the main byproduct and the recovered starting material.
- 6) Based on the NMR spectra reported in the supporting information, compounds 3 appear to be frequently isolated as mixture of rotamers. Authors should mention this aspect in the main document. Proton and carbon NMR analyses could be performed at different temperatures to try and obtain better resolved spectra for these compounds.
- 7) The heading of the final section in the manuscript should be "conclusions" instead of "discussion".

8) In the supporting information, HRMS data for compound 4a should be carefully checked. When describing HRMS data, the term “requires” should be replaced with “calculated” (calcd).

9) In the SI, the authors should describe the synthesis of the CPA catalysts employed to perform the transformations, or clearly mention the supplier if these compounds are commercially available.

10) Authors should mention in the supporting information the method employed to obtain the racemic products 3 used as references for the HPLC analyses.

Reviewer #2 (Remarks to the Author):

The authors describe here a novel kinetic resolution (KR) method for amido-PCPs, by taking advantage of a method they have applied recently on other substrates such as propargylic anilines (ref 40), BINAM and NOBIN (ref 41) and hydroquinolines (ref 39), consisting in chiral phosphoric acid (CPA) catalyzed asymmetric C-H amination with azodicarboxylates. In this paper the authors optimize first the CPA and reaction conditions, and then they investigated the scope of the method which is really broad. Mechanistic studies point out towards a sequential triazane formation and N[1,5]-rearrangement process.

The results are important, useful for the community of PCP and the manuscript is very clear and well organized. I have however a doubt about its suitability for Nature Commun., since it seems to me that such a methodology work, although very well conducted, lack somehow of novelty. The authors published already a series of very recent papers (refs 39-41) in high profile journals, by introducing this KR method and applying it to different substrates.

I have noticed several corrections to be done:

1) In the Abstract, the CPA term in “CPA-catalyzed” should be defined.

2) In Fig. 1, the name of the compounds should be carefully checked. The first one, for example!

3) In Table 1, what is the difference between A4 and A7? What is BIONOL with respect to BINOL?

4) In Fig. 4c SCl_2 should be SCCl_2 (thiophosgene).

5) In the X-ray Table for (S)-1f the value of the Flack parameter should be included.

Reviewer #3 (Remarks to the Author):

In this manuscript, Yang and coworkers described a CPA-catalyzed efficient and versatile kinetic resolution reaction, which could construct various substituted amido[2.2]paracyclophanes by asymmetric amination. The reaction mechanism was studied by several control experiments, and a novel reaction mechanism for the electrophilic aromatic C-H amination, which proceeded through sequential triazane formation and N[1,5]-rearrangement was proposed. And various derivations and application of the chiral products showed the potential value of this reaction.

In conclusion, this is a nice piece of well-executed work which meets the requirements of novelty and originality necessary for publication in Nature Communications. Only few organocatalysed reactions allow to stereoselectively construct chiral PCP derivatives. This methodology is very attractive and complementary. Consequently I give my support for publication in Nature Communications after addressing these points.

1) "4A" should be replaced by "4Å" and a space is required between number and unit.

2) In the Fig. 2, various substituted amido[2.2]paracyclophanes have good reaction effect. But there are no ortho- or meta-substituted substrates for aniline, if these compounds been used, can the corresponding products be obtained?

3) The authors indicate that the CPA-catalyzed direct addition of the amido group to azodicarboxylate (formation of triazanes 8) was believed to be the stereodetermining step. In that case enantiomer 3 could be constructed from enantiomer 8 catalyzed by racemic phosphoric acid?

4) The picture of HPLC provided is not standard, please reprocess.

5) Many peaks in the NMR spectrum are too low, making it impossible to determine the purity of the product. Please readjust it. And integrate the nuclear magnetism correctly.

Reviewer #1

The manuscript of Yang and co-workers describes a highly efficient kinetic resolution of differently substituted amido[2.2]paracyclophanes. This versatile transformation proceeds via an asymmetric electrophilic amination promoted by chiral phosphoric acid catalysts. After the optimization of the reaction conditions, the kinetic resolution protocol was readily applied to a wide variety of racemic substrates. A valuable desymmetrization of a centrosymmetric diamido pCp was also achieved following this strategy. Several control experiments have been conducted by the authors to propose a plausible reaction mechanism. Finally, the utility of the enantioenriched products obtained using this method was showcased by performing useful derivatization reactions. High added value compounds were easily obtained, including an example of a promising organocatalysts.

The work is clearly designed, and the conclusions are consistent with the experimental data. The supporting information appears complete. The proposed transformation grants a straightforward access to enantioenriched nitrogen-containing pCps that can serve as valuable synthetic intermediate for a variety of applications ranging from asymmetric catalysis to material sciences. I believe that the results of this study will be relevant to a broad readership of scientists. I therefore think that this manuscript should be accepted for publication in Nature Communications after addressing the following minor remarks.

Response: We thank the anonymous reviewers for careful reading and constructive comments, which has strengthened this work.

1) The manuscript is well-structured, but some typos can be found in the text and a few sentences are hard to understand. English needs to be polished.

Response: The language of this manuscript has been carefully examined and polished.

2) When describing the optimization of the reaction conditions, authors should comment on the solubility of the starting materials and products in the solvents employed to perform the kinetic resolutions.

Response: In the model reaction presented in Table 1, both the starting material **1a** and the product **3a** exhibited good solubility.

3) Authors should more clearly describe the advantages and disadvantages of employing CCl₄ as a solvent instead of toluene.

Response: The advantage of employing CCl₄ as the solvent is that the reaction conducted in CCl₄ exhibited the highest reaction rate, which would enable a decrease in both the reaction temperature and the necessary amount of azodicarboxylate. However, the disadvantage is the toxicity associated with CCl₄.

4) Screening of different temperatures for the reaction performed in toluene may be interesting and could be added to the optimization table.

Response: Recognizing the toxicity associated with CCl₄, the reaction was also explored in toluene at lower temperatures (entries 18-19). Although this resulted in slightly diminished KR performance, it nonetheless offers an alternative set of conditions for the KR of

amido[2.2]paracyclophanes.

5) In figure 2, when describing the desymmetrization reaction of the meso compound 4a, authors should report more precisely the yield of the main byproduct and the recovered starting material.

Response: The starting material was obtained in a yield of 23%, while the exact yield of the achiral di-amination byproduct could not be accurately determined due to its mixture with CPA catalysts and other unidentified compounds.

6) Based on the NMR spectra reported in the supporting information, compounds 3 appear to be frequently isolated as mixture of rotamers. Authors should mention this aspect in the main document. Proton and carbon NMR analyses could be performed at different temperatures to try and obtain better resolved spectra for these compounds.

Response: The varied-temperature NMR experiment was conducted for compound 3a (refer to pages 321-322 in the Supporting Information), which resulted in significantly improved clarity of the ¹H NMR spectra, suggesting the presence of a rotamer mixture at room temperature.

7) The heading of the final section in the manuscript should be “conclusions” instead of “discussion” .

Response: The heading of the final section in the manuscript has been changed to “conclusions”.

8) In the supporting information, HRMS data for compound 4a should be carefully checked. When describing HRMS data, the term “requires” should be replaced with “calculated” (calcd).

Response: The term “requires” has been replaced with “calculated”.

9) In the SI, the authors should describe the synthesis of the CPA catalysts employed to perform the transformations, or clearly mention the supplier if these compounds are commercially available.

Response: The CPA catalysts employed in this work are purchased from bidepharmatech. This statement has been added in the General Information section in the Supporting Information.

10) Authors should mention in the supporting information the method employed to obtain the racemic products 3 used as references for the HPLC analyses.

Response: The racemic products were afforded by using racemic phosphoric acid A4 as a catalyst.

Reviewer #2 (Remarks to the Author):

The authors describe here a novel kinetic resolution (KR) method for amido-PCPs, by taking advantage of a method they have applied recently on other substrates such as propargylic anilines (ref 40), BINAM and NOBIN (ref 41) and hydroquinolines (ref 39), consisting in chiral phosphoric acid (CPA) catalyzed asymmetric C-H amination with azodicarboxylates. In this paper the authors optimize first the CPA and reaction conditions, and then they investigated the scope of the method which is really broad. Mechanistic studies point out towards a sequential triazane

formation and N[1,5]-rearrangement process.

The results are important, useful for the community of PCP and the manuscript is very clear and well organized. I have however a doubt about its suitability for Nature Commun., since it seems to me that such a methodology work, although very well conducted, lack somehow of novelty. The authors published already a series of very recent papers (refs 39-41) in high profile journals, by introducing this KR method and applying it to different substrates.

Response: We thank the anonymous reviewers for careful reading and constructive comments, which has strengthened this work.

However, we respectfully disagree with one reviewer's assessment regarding the novelty of our work. While similar approaches have been effectively employed in the kinetic resolution of arylamines with central and axial chirality, this method's application to the kinetic resolution of amido[2.2]paracyclophanes is unparalleled and unforeseen. Furthermore, we discovered a novel reaction mechanism for the electrophilic aromatic C-H amination, involving sequential triazane formation and N[1,5]-rearrangement.

I have noticed several corrections to be done:

1) In the Abstract, the CPA term in “CPA-catalyzed” should be defined.

Response: The term of CPA in the abstract was defined as “chiral phosphoric acid”.

2) In Fig. 1, the name of the compounds should be carefully checked. The first one, for example!

Response: The name of the first compound in Fig. 1 was revised.

3) In Table 1, what is the difference between A4 and A7? What is BIONOL with respect to BINOL?

Response: The difference between **A4** and **A7** is that **A4** possess the chiral BINOL scaffold, while **A7** possess the chiral H8-BINOL scaffold.

The term of “BIONOL” has been revised to “BINOL”.

4) In Fig. 4c SCI2 should be SCCI2 (thiophosgene).

Response: “SCI2” has been modified to “SCCI2” in Fig. 4c.

5) In the X-ray Table for (S)-1f the value of the Flack parameter should be included.

Response: The value of the Flack parameter for (S)-1f has been added.

Reviewer #3 (Remarks to the Author):

In this manuscript, Yang and coworkers described a CPA-catalyzed efficient and versatile kinetic resolution reaction, which could construct various substituted amido[2.2]paracyclophanes by asymmetric amination. The reaction mechanism was studied by several control experiments, and a novel reaction mechanism for the electrophilic aromatic C-H amination, which proceeded through sequential triazane formation and N[1,5]-rearrangement was proposed. And various derivations and application of the chiral products showed the potential value of this reaction.

In conclusion, this is a nice piece of well-executed work which meets the requirements of novelty and originality necessary for publication in Nature Communications. Only few organocatalysed

reactions allow to stereoselectively construct chiral PCP derivatives. This methodology is very attractive and complementary. Consequently I give my support for publication in Nature Communications after addressing these points.

Response: We thank the anonymous reviewers for careful reading and constructive comments, which has strengthened this work.

1) “4A” should be replaced by “4 Å” and a space is required between number and unit.

Response: “4A” has been revised to “4 Å”.

2) In the Fig. 2, various substituted amido[2.2]paracyclophanes have good reaction effect. But there are no ortho- or meta-substituted substrates for aniline, if these compounds been used, can the corresponding products be obtained?

Response: Synthesizing the *ortho*- or *meta*-substituted amido[2.2]paracyclophanes poses significant challenges. For instance, when various *ortho*-halogenation conditions were applied to **1a**, only the *para*- and *ortho,para*-disubstituted products were obtained. Furthermore, attempts to selectively remove the less sterically hindered *para*-halide did not yield the desired *ortho*-substituted amido[2.2]paracyclophanes.

3) The authors indicate that the CPA-catalyzed direct addition of the amido group to azodicarboxylate (formation of triazanes **8**) was believed to be the stereodetermining step. In that case enantiomer **3** could be constructed from enantiomer **8** catalyzed by racemic phosphoric acid?

Response: The enantioenriched triazane (*Sp*)-**8x** (94% ee) was readily converted into the CH-amination product (*Sp*)-**3x** (93% ee) enabled by a racemic catalyst (see page 86 in Supporting Information for details), which provided additional evidence that the formation of the triazane served as the stereodetermining step.

4) The picture of HPLC provided is not standard, please reprocess.

Response: The full HPLC spectra have been added.

5) Many peaks in the NMR spectrum are too low, making it impossible to determine the purity of the product. Please readjust it. And integrate the nuclear magnetism correctly.

Response: Some of the NMR spectra of the products **3** have been modified. However, it is important to highlight that the seemingly lower quality of certain NMR spectra was attributed to the presence of rotamers in the amination products at room temperature. For instance, a varied-temperature NMR experiment was performed on compound **3a** (see pages 321-322 in the Supporting Information), leading to a notable enhancement in the clarity of the ^1H NMR spectra.

REVIEWERS' COMMENTS

Reviewer #3 (Remarks to the Author):

In this response, Yang and coworkers have perfectly addressed the queries raised by the reviewers, and therefore I consider the manuscript suitable for publication in Nature Communications.

Reviewer #3 (Remarks to the Author):

In this response, Yang and coworkers have perfectly addressed the queries raised by the reviewers, and therefore I consider the manuscript suitable for publication in Nature Communications.

Response: We thank the anonymous reviewers for careful reading and constructive comments, which has strengthened this work.